# Fluctuating Asymmetry in *Asteriscii* Otoliths of Common Carp (*Cyprinus carpio*) Collected from Three Localities in Iraqi Rivers Linked to Environmental Factors

Laith Jawad [1] and Kélig Mahé [2,*]

1   School of Environmental and Animal Health Sciences, Unitec Institute of Technology, 139 Carrington Road, Mt Albert, Auckland 1025, New Zealand; laith_jawad@hotmail.com
2   IFREMER, Fisheries Laboratory, 150 Quai Gambetta, BP 699, 62321 Boulogne-sur-Mer, France
*   Correspondence: kelig.mahe@ifremer.fr; Tel.: +33-321-995-602

**Abstract:** Otoliths, calcified structures in the inner ears, are used to estimate fish age, and their shape is an efficient fish stock identification tool. Otoliths are thus very important for the management and assessment of commercial stocks. However, most studies have used left or right otoliths, chosen arbitrarily without evaluation of the difference between these otoliths. In this study, the *asteriscii* otoliths from 263 common carp (*Cyprinus carpio*; Linnaeus, 1758) were sampled in three Iraqi rivers to test the potential asymmetry and the geographical effect on otolith growth from three measurements (length, width and weight), and on shape from two shape indices (ellipticity and form-factor). Among all *asteriscii* otolith features, there was significant fluctuating asymmetry between fish length and every otolith descriptor. At one fish length, the size and/or the shape of otoliths could be different between two individuals and/or between left and right *asteriscii* otoliths for the same individual. Moreover, the relationship between fish length and otolith shape/growth was significantly dependent on the studied geographical area and, more especially, the environmental effects as the water temperature and pH. Finally, the relationships between fish length and otolith shape indices showed that the otolith evolves into the elliptical shape during the life of the fish. To use the otolith shape, it is essential to take into account the developmental stage of individuals to integrate the ontogenetic effect. Our results highlight the importance of verifying potential otolith asymmetry, especially for the *asteriscii* otoliths (lagenar otoliths) before their use in fisheries research.

**Keywords:** *asteriscus*; side effect; growth; otolith shape; geographical effect; temperature effect; pH

## 1. Introduction

Otoliths, calcified structures in the inner ears [1,2], are used to identify species in taxonomic or phylogenic studies and to collect age data for management and assessment of stocks. The otoliths grow throughout the fish life and are metabolically inert [3], and their shape (i.e., their outline resulting from genetic, environmental and ontogenetic effects) is used as an efficient tool to recognize the species at the interspecific level and to identify the fish stocks at the intraspecific level (to see the Stock Identification Methods Working Group of which identifies all studies each year; SIMWG). Each head side (i.e., left and right inner ear), however, has three pairs of otoliths (*sagittae*, *asteriscii* and *lapilli*) [4]. Among the three different otoliths, the sagittal otolith is usually used due to its larger size and easier removal [5]. The *asteriscii* (lagenar otoliths) are the smallest otolith in marine species and so rarely used. Conversely, *asteriscus* otoliths are the most frequently used otoliths in Cypriniforms species such as common carp [6]. For these species, *asteriscii* are the largest of the three otolith pairs [6]. Several growth studies focused on the common carp have used *asteriscii* otoliths [6–9]; however, no study has focused on the *astericus* shape for this species.

Since 1990, when the first report on the asymmetry in fishes was published on fishes of Iraq [10], few scientific publications have appeared on the phenomenon of asymmetry

in freshwater fish species. In total, there are four publications concerning the Iraqi waters reporting asymmetry in four fish species (*Heteropneustes fossilis*, Bloch, 1794; *Mystus pelusius*, Solander, 1794; *Planiliza abu*, Heckel, 1843; and *Parasilurus triostegus*, Heckel, 1843) related to different fish characters [10–14]. There are similarly few reports for the marine fish species of Iraq: the first publication on the asymmetry of this group was in 2020, with six reports studying otolith asymmetry in six marine species. The present study focuses on potential asymmetry and the geographical effects on otoliths growth and shape in common carp *Cyprinus carpio*, a freshwater species of Iraq. A recent study using the otolith shape to identify the stock structure of the bogue (*Boops boops*; Linnaeus, 1758) in the Mediterranean Sea showed that the significant asymmetry could modify the boundaries of stocks according to the use of the left or right otolith [15]. Moreover, this study showed that this significant asymmetry could be due to environmental differences. Consequently, the aims of this study are to compare the fluctuations in asymmetry of the *asteriscus* in the carp, *Cyprinus carpio,* and to identify if this asymmetry could be a possible response to the environmental variables in several rivers in Iraq.

## 2. Materials and Methods

### 2.1. Sample Collection

A total of 263 individual carp were collected in May–July 2021 from three different locations on the Euphrates (Nasiriya City; 31°2′28″ N, 46°14′45″ E; $n = 90$), Tigris (Amarah City; 31°51′25″ N, 47°8′15″ E; $n = 89$) and Shatt al-Arab (Basrah City; 30°33′41″ N, 47°47′41″ E; $n = 84$) rivers (Figure 1; Supplementary Table S1). All individuals were from commercial catches and macroscopic observation of the gonads did not identify sex with a good accuracy.

### 2.2. Study Areas

The Euphrates is the longest river in Western Asia (around 3000 km; [16–19]). At Nasiriya City, where the common carp specimens were collected for this study, the water of the Euphrates river is seasonally variable in temperature, salinity and hydrogen ion concentration. Water temperature ranged from 11 °C in January to 34 °C in July and pH values fluctuated from 7.12 in August to 8.43 in December [16].

The Tigris is 1750 km long, rising in the Taurus Mountains of eastern Turkey, about 25 km southeast of Elazig city and about 30 km from the headwaters of the Euphrates [20]. The water temperature of the Tigris river in Mysan Province varied between 15 °C and 25 °C and pH values from 7.05 to 7.8 [21].

The Shatt al-Arab river, around 200 km long, is formed at the confluence of the Euphrates and Tigris rivers in al-Qurnah City in southern Iraq [22]. Currently, the river depends mainly on freshwater flow from the Tigris river [23,24]. The water flow in the Shatt Al-Arab river is affected by the tidal phenomenon of the Arabian Gulf, which has a semidiurnal pattern [25,26]. The recorded water temperature of the Shatt al-Arab river at Basrah City varied between 19 and 32 °C and the pH values between 7.86 and 8.07 [27,28].

### 2.3. Morphometrical Analysis

The commercial catches were sampled directly at the laboratory to limit the storage bias on the fish and otolith data. The first step was to measure the total length (TL $\pm$ 0.1 cm) of fish, then extract and clean both *asteriscii* otoliths. To describe the otolith shape, only univariate data as size parameters and shape factors were used. *Asteriscii* Otolith weight ($O_{WEIGHT}$) was measured using a digital balance to the nearest 0.0001 g (Sartorius Precision Balance Entris; Model BCE6231-1CFR, Sartorius Lab Instruments GmbH & Co. KG Otto-Brenner-Strasse 20 37079 Goettingen, Germany). Otolith images were captured using a camera (Efix 07-45x, 13MP, Alibaba.com, China (accessed on 28 March 2022)) with a stereomicroscope. *Asteriscii* Otolith length ($O_{LENGTH}$, mm) and width ($O_{WIDTH}$, mm) were taken using image processing systems (detailed descriptions in Figure 2). Size parameters (otolith length, width and weight) are measurements linked directly to otolith size, and

are linked to otolith growth. Conversely, shape indices are dimensionless (and thus independent from otolith size) measures of otolith morphology similarity (i.e., otolith shape), compared with ideal geometric shapes calculated using size features. Two shape indices were used: ellipticity (to quantify similarity to an ellipse) and aspect-ratio (to compare to a rectangle) [29].

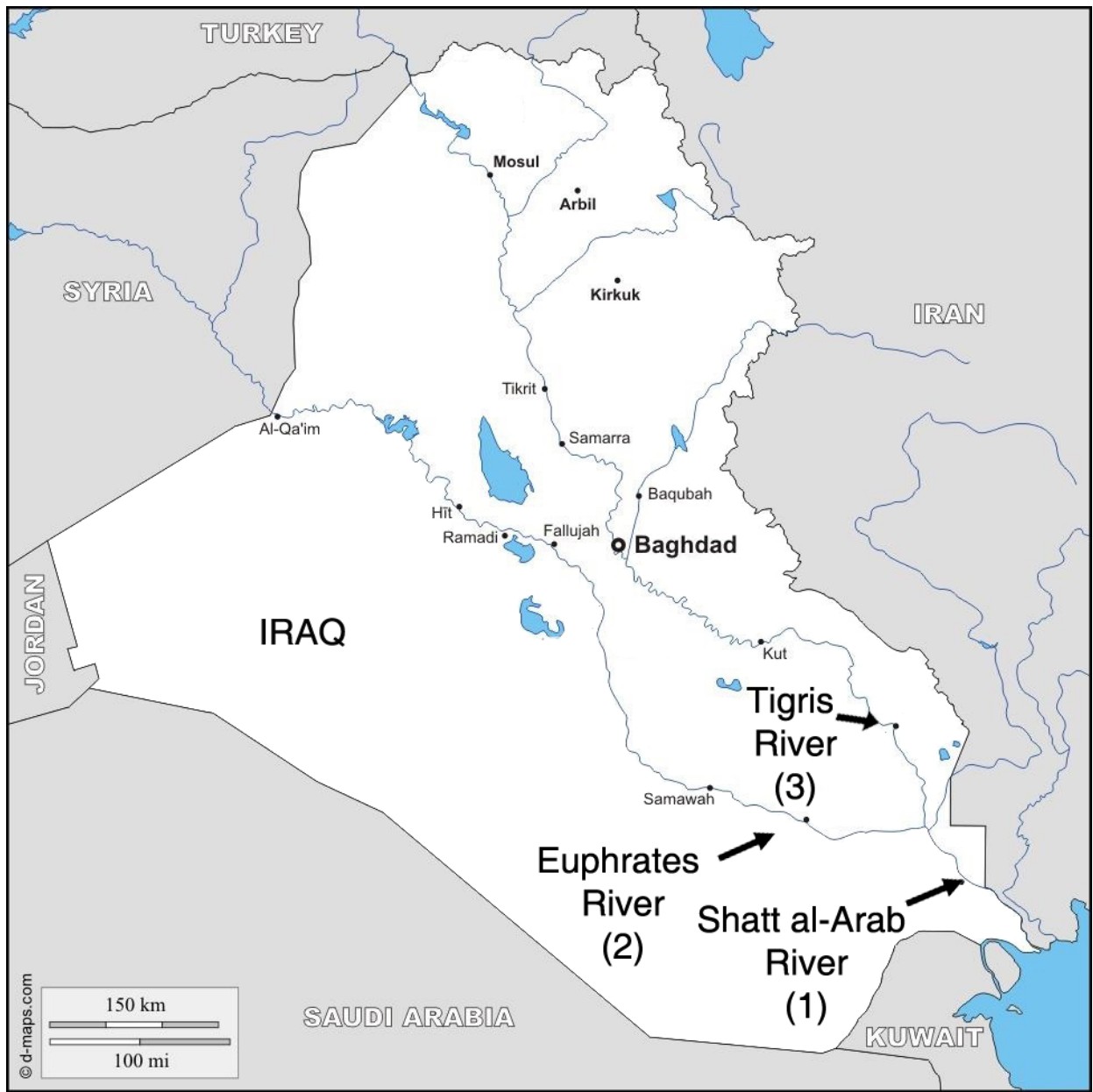

**Figure 1.** Sampling locations where common carp (*Cyprinus carpio*) were collected: the Shatt al-Arab (1), Euphrates (2) and Tigris (3) rivers.

$$\text{Ellipticity} = \frac{\text{OLENGTH} - \text{OWIDTH}}{\text{OLENGTH} + \text{OWIDTH}} \qquad (1)$$

$$\text{Aspect} - \text{Ratio} = \frac{\text{OLENGTH}}{\text{OWIDTH}} \qquad (2)$$

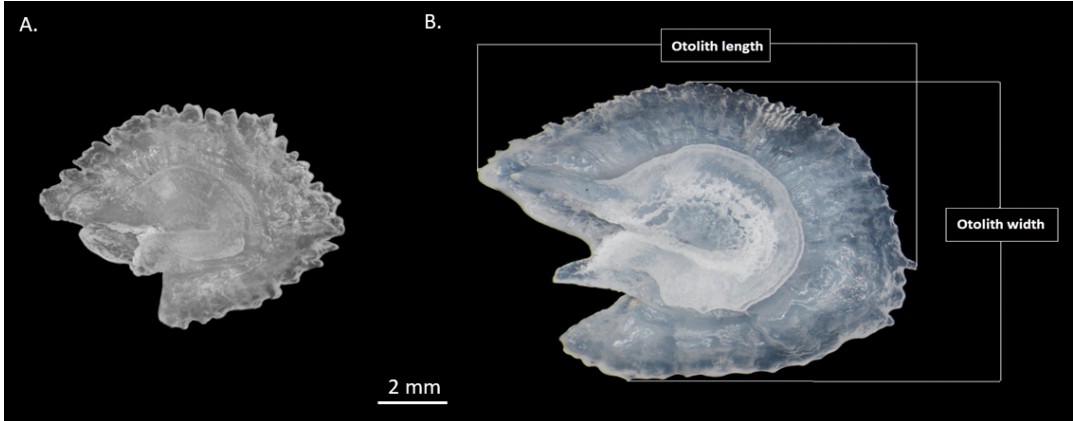

**Figure 2.** *Asteriscii* otoliths of *Cyprinus carpio* illustrating different features of the otolith measurements for two life stages ((**A**): small individual, TL = 155 mm; (**B**): large individual, TL = 380 mm).

The relationship between fish total length and each *asteriscii* otolith feature (otolith length, width, weight, ellipticity and aspect-ratio) according to the geographical position (River) and head side (Side) was modelled according to Equation (3):

$$TL \sim Otolith\ feature + Otolith\ feature{:}River + Otolith\ feature{:}Side + Otolith\ feature{:}River{:}Side \quad (3)$$

One generalized linear model was performed by each otolith feature (otolith length, width, weight, ellipticity and aspect-ratio). The geographical (*Asteriscii* Otolith feature:River) and side (*Asteriscii* Otolith feature:Side) effects and their interaction (Otolith feature:River:Side) were tested by the relationships between fish length and otolith feature. A second model was applied to evaluate the environmental effects (water temperature: T °C and acidification: pH) on the relationship between fish size and all otolith features according to the head side (Equation (4)). The environmental data were extracted from the literature for each location [16,21,27,28]. To test the environmental parameters, we used three values by factor with the annual mean value, the minimum and the maximum values registered by area.

$$TL \sim Otolith\ feature{:}T\ ^{\circ}C{:}Side + Otolith\ feature{:}pH{:}Side \quad (4)$$

The normality and the homoscedasticity of the residuals were assessed by visual inspection of diagnostic plots. The significance of explanatory variables was tested by likelihood ratio tests between nested models, while respecting the marginality of the effects (type 2 tests [29]) that are supposed to follow an $\chi^2$ distribution under the null hypothesis, while correcting for multiple comparisons using a Bonferroni procedure. Statistical analyses were performed in the R statistical environment [30], using car [31], sp. [32], HH [33], vegan [34] and ggplot2 [35] packages.

## 3. Results

There were some differences between left and right *asteriscii* otoliths from the same individuals with the right otolith being bigger (i.e., otolith length and width) than the left otolith (Figure 3). Analysis of the relationships between fish length and each otolith feature (three size parameters and two shape indices) showed that there was a significant relationship between only two otolith size descriptors (length and width) with the total length of fish (column "TL", Table 1). For three other otoliths features, the relationship with the total length was not significant (Figure 4).

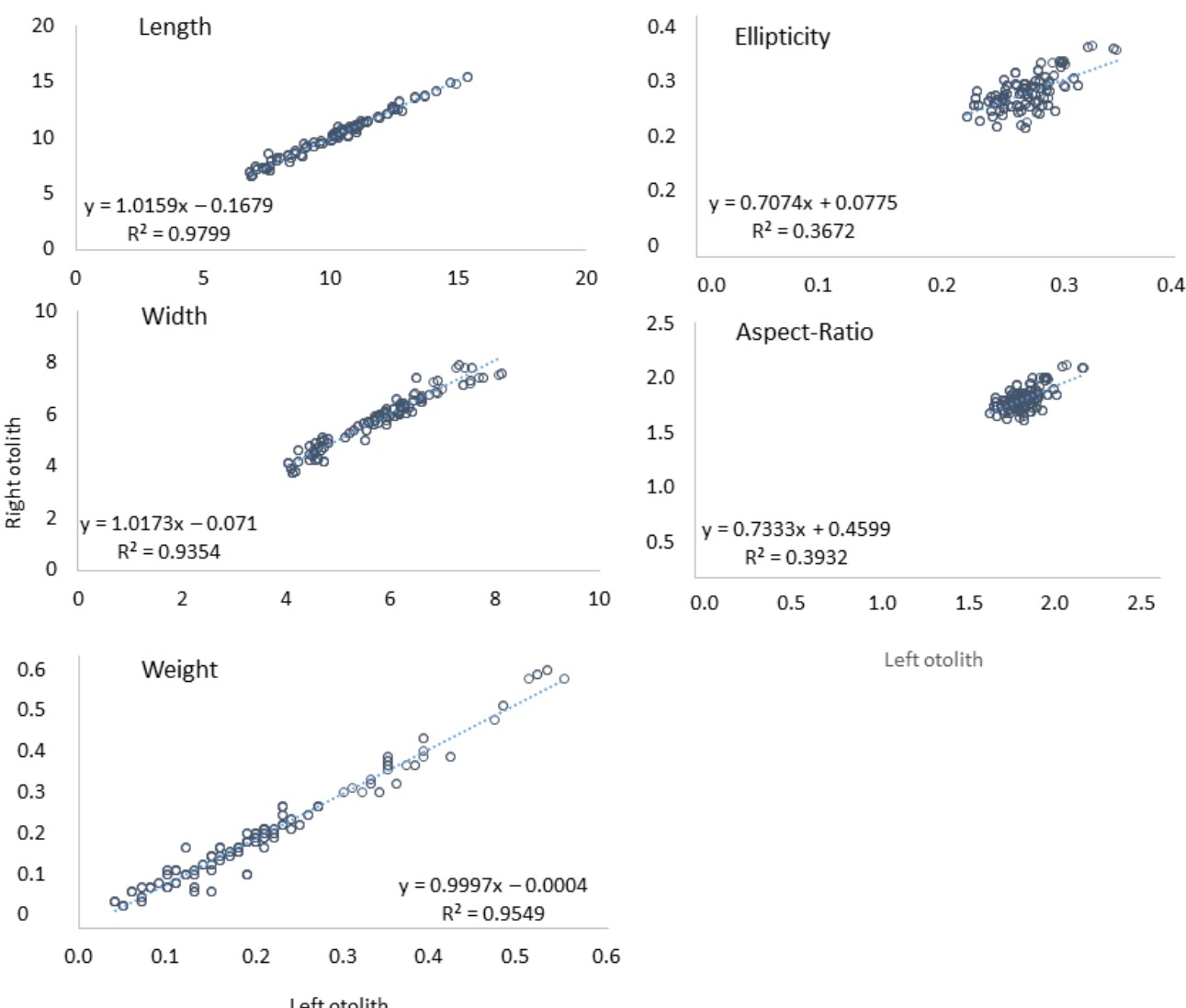

**Figure 3.** Difference between left and right otoliths of *Cyprinus carpio* for each otolith measurement.

**Table 1.** *p*-values of generalized linear models for the relationship between fish length and each otolith variable for *Cyprinus carpio* collected from three localities in Iraqi rivers with by geographical and Side effects (observed from the interaction between Otolith feature and river or side in the GLM: Otolith feature: River/Side). The environmental effects on the relationship between fish length and each otolith variable (Otolith feature: T °C/Side and Otolith feature: pH/Side) were tested with three different values (mean, minimal and maximum). Grey cases show significant effects ($p < 0.05$).

| Otolith Descriptor | TL | Geographical Effect | Side Effect | Geographical Effect/Side Effect | T°C Effect/Side Effect | | | pH Effect/Side Effect | | |
|---|---|---|---|---|---|---|---|---|---|---|
| | | | | | mean | min | max | mean | min | max |
| $O_{LENGTH}$ | 0.004 | 0.213 | 0.005 | 0.331 | 0.011 | 0.053 | 0.434 | 0.004 | 0.005 | 0.222 |
| $O_{WEIGHT}$ | 0.051 | 0.278 | <0.001 | 0.187 | 0.059 | 0.212 | 0.529 | 0.029 | 0.048 | 0.307 |
| $O_{WIDTH}$ | 0.005 | 0.064 | 0.027 | 0.058 | 0.007 | 0.042 | 0.416 | 0.003 | 0.010 | 0.283 |
| Ellipticity | 0.326 | 0.008 | 0.014 | 0.029 | <0.001 | 0.009 | 0.265 | <0.001 | 0.003 | 0.151 |
| AR | 0.912 | 0.002 | 0.032 | 0.043 | <0.001 | 0.005 | 0.236 | <0.001 | 0.006 | 0.154 |

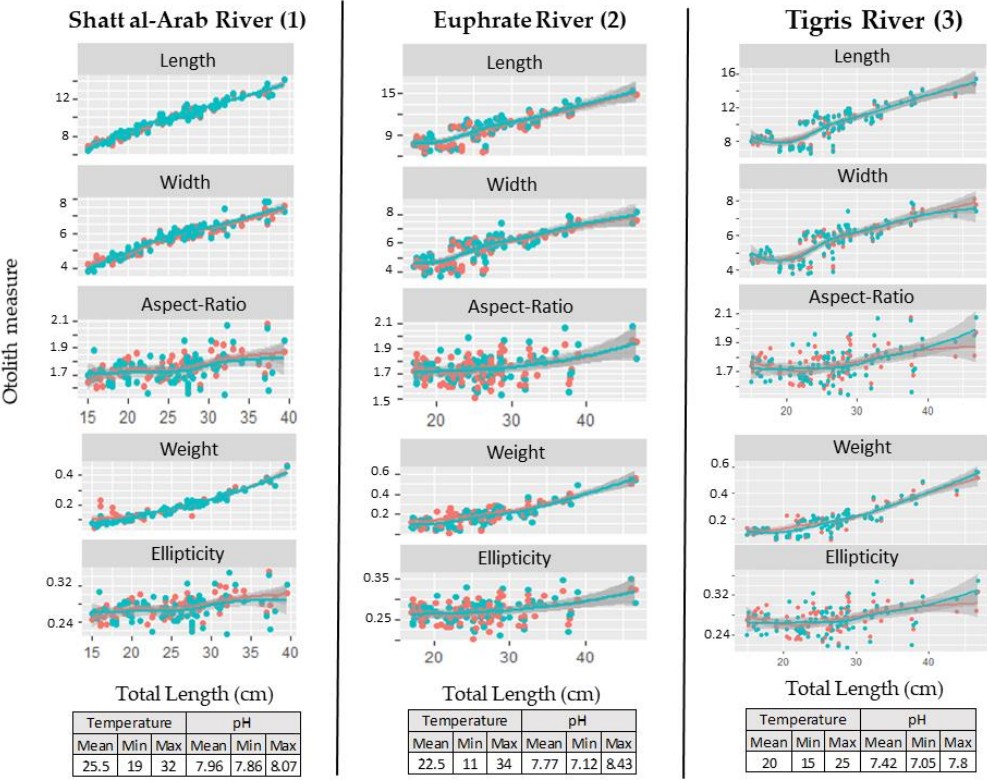

**Figure 4.** Relationships between fish length and *asteriscii* otolith morphological features (red points and line = left otolith, and green points and line = right otolith) according to the head side of *Cyprinus carpio* for each Iraqi River (for each river, the environmental data are presented).

Among the five *asteriscii* otolith features (size parameters or shape indices), there is always a significant effect of the head side on the relationship between fish length and each otolith descriptor (observed from the interaction between otolith feature and area or side in the GLM: Otolith feature: River/Side; Table 1). The relationship between fish length and otolith shape (i.e., two shape indices) was significantly dependent on the studied geographical area (i.e., river). The data collected in Tigris river presented the highest level of the fluctuating asymmetry in *asteriscii* otoliths of common carp collected from three localities in Iraqi Rivers. The relationship between fish and otolith growth was not directly linked to the location of sampling (Table 1). Finally, this generalized linear model showed that the relationship between fish length and otolith shape described by ellipticity and aspect-ratio is significantly different between the right and left otoliths, and this result was also linked to the location of sampling. Moreover, two mainly environmental factors were tested on the side effect applied to the relationship between fish length and each otolith descriptor. The results showed the interaction between both water temperature and pH and head side on the relationship between fish length and each otolith descriptor, especially with the minimum and mean values of these factors (Table 1). The Tigris river, which presented the lowest value of water pH and temperature, was the location with the highest level of the fluctuating asymmetry in *asteriscii* otoliths.

The significant correlation between fish total length and otolith growth showed that the following factors of otolith size also increased with an increase in total length: length, width and weight (Figure 4). The relationship between body length and otolith shape indices showed that the otolith shape evolves into a rectangular or elliptical shape during the life of the fish. The difference between the shape of the left and right otoliths in the same fish increased mainly in large individuals over 40 cm. This asymmetry is explained mainly by the width of the *asteriscii* otolith (Figure 4).

## 4. Discussion

The morphogenesis of the otolith is the result of a combination of exogenous (environment), endogenous (genetics) and developmental (ontogeny) factors [36,37]. Thus, the variability of these different factors and the biological scale at which they are expressed will determine the variation in the growth and shape of the otolith. Environmental factors that influence the morphogenesis of the otolith are separated into two groups: firstly, there are abiotic factors with temperature [38,39], pH [40], the depth of the water in which the fish live [41], the nature of the substrate [42] and salinity [43]. The growth and the morphogenesis of otoliths during the early life stages of fish are due to ontogenetic effects, but they also temperature-dependent [38]. In the same way, water acidification can also alter otolith shape [40]. For several species, individuals exposed to high $pCO_2$ had a larger otolith area and maximum length compared with controls; the increases were larger than could be explained by an increase in $CaCO_3$ precipitation in the otoliths driven by the modification of the pH regulation in the endolymph [44–47]. Our study corroborated that the water temperature and pH modified the relationship between fish length and each otolith measure (i.e., otolith length and width) and shape (i.e., ellipticity and aspect-ratio). Secondly, biotic factors such as the quantity [48] and specific composition [49] of the food available to the fish also affect the morphogenesis of the otolith. Genetic variability is also a strong factor influencing this morphogenesis process [50–55]. Ontogeny is also a factor generating shape variability, which can be reflected in effects of sex, age [53], body size [48] or the stage of sexual maturity [42,50,53] of the individual. The development of the otolith is spatially heterogeneous. The otoliths are paired structures present in the right and left inner ears, and an asymmetry between the two otoliths, or fluctuating asymmetry (FA), can be measured. This fluctuating asymmetry is related to the developmental trajectory of the otoliths, which is itself guided by developmental regulatory processes, such as the evolutionary canalization [54]. Fluctuating asymmetry may be related to developmental instability and thus provide an indicator measure of stress or micro-environmental variability [55–58]. In particular, Lemberget and McCormick [56] identify FA as an indicator of fish health as this type of asymmetry can directly affect the balance and hearing performance of the fish. This asymmetry can also be a functional adaptation to the environment. Our study showed the significant interaction between the level of asymmetry and the environmental factors (water temperature and pH). The acidification and the lowest temperature (i.e., lowest value of pH and temperature of water) observed in the Tigris river showed the highest level of asymmetry registered in Iraqi Rivers. As the trends of pH and temperature between the three locations were the same direction, it was not possible to dissociate the acidification and the temperature effects. The asymmetry can be measured in the otolith growth from the morphological descriptors (i.e., the relationship between fish length and otolith descriptors such as weight, length and width) and/or the otolith shape from univariate (i.e., Shape index) or multivariate (i.e., Elliptic Fourier Descriptors) measures. Otolith mass asymmetry is the most commonly used measure to compare the growth between left and right otoliths. Our results showed that there was a significant asymmetry in otolith growth of common carp (*Cyprinus carpio*) observed from all otolith descriptors (i.e., length, width and weight). This asymmetry corroborates that observed in previous studies on other marine and freshwater species [49,58–67]. In contrast, one study focused on Cyprinid fishes from Turkish inland waters observed no difference in the relationship between otolith descriptors and fish length according to the side effect [68]. However, the previous study on common carp (*Cyprinus carpio*) estimated that this species showed fluctuating asymmetry in otolith weight [69].

Otolith mass asymmetry has been used as an indicator to evaluate condition between several habitats [70], in particular, as a consequence of environmental stress, human activities, genetic disposition and a combination of these factors [71]. Our results showed that there was a significant asymmetry in otolith shape of common carp (*Cyprinus carpio*) observed from two shape indices (i.e., ellipticity and form factor) with increased asymmetry with size. During the life of the fish, the ontogenetic effect on the shape of the otolith de-

creased, unlike the environmental effects (temperature, food, etc. [36,39,42]). Consequently, this observed trend showed that asymmetry could be the result of the environmental conditions in each river. The water temperature and pH showed that the fluctuating asymmetry could be linked significantly to these two environmental factors. Our results corroborate those of previous studies on other marine and freshwater species [54,72–75].

## 5. Conclusions

This study showed that the observed asymmetry level was significantly linked to the geographical area, with the otolith shape being linked only to the environmental effect (i.e., water temperature and pH). In the climate change context, the temperature and pH are mainly environmental factors, which will be modified. Consequently, in the future, the asymmetry level may change and alter the balance and hearing performance of the fish. Sagittal otoliths, such as *asteriscii*, may also show significant asymmetry. Moreover, the experimental approach should be used to better understand the factors controlling otolith shape, integrating the ontogenetic effect, as has been undertaken for other species. Finally, this is the first study showing that it is necessary to estimate the level of asymmetry between otolith pairs to limit the potential bias due to a difference between left and right sides when using otoliths, and that the asymmetry could be linked directly to environment factors such as temperature and pH.

**Supplementary Materials:** The following are available online at https://www.mdpi.com/article/10.3390/fishes7020091/s1, Table S1: Size ranges of *Cyprinus carpio* collected from three localities in Iraq Rivers by head side (sample size, characteristics of fish length and each otolith variable).

**Author Contributions:** L.J. and K.M. designed the research; L.J. realized the sampling. K.M. performed the statistical analyses. All authors provided input for the results and discussion and wrote the paper. All authors provided critical comments and were involved in the writing of the manuscript. All authors accepted the final version of the manuscript. All authors have read and agreed to the published version of the manuscript.

**Funding:** This research received no external funding.

**Institutional Review Board Statement:** Ethical review and approval were waived for this study, because the fish were obtained from fisheries and were already dead when the otoliths were extracted.

**Data Availability Statement:** Data Availability Statements in section "MDPI Research Data Policies" at https://www.mdpi.com/ethics (accessed on 28 March 2022).

**Acknowledgments:** We would like to express our gratitude to all people involved in the collection of samples required in this study. We would especially thank Kirsteen MacKenzie for her valuable help in editing this manuscript.

**Conflicts of Interest:** The authors declare no conflict of interest.

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
