# Peer review of "Fluctuating Asymmetry in Asteriscii Otoliths of Common Carp (Cyprinus carpio) Collected from Three Localities in Iraqi Rivers Linked to Environmental Factors"

_fishes, doi:10.3390/fishes7020091_

Round 1
Reviewer 1 Report
General suggestions:
Please indicate species authorities each first time a species is mentioned, e.g. Cyprinus carpio (Linnaeus, 1758)
Otolith is a too general term, please use the specific term (asteriscus/ii) in indicating this structure
Introduction:
- lines 29-30 Otoliths are also use in taxonomy, and phylogeny, thus, are not primarily used for fisheries sciences
- please wide the introduction section regarding shape analyses ("Intra‐ and interspecific variability among congeneric Pagellus otoliths" (SciRep, 2021), "Otolith Analyses Highlight Morpho-Functional Differences of Three Species of Mullet (Mugilidae) from Transitional Water" (Sustainability, 2022), "ShapeR: An R package to study otolith shape variation among fish populations" PLoS ONE 10, 1–12 (2015)
Material and methods:
- it is preferable to present "samples collection" before describing "study areas"
- in describing the sampling point please be sure to indicate geographical coordinates both in the text and in preparing the map reported in figure 1. Maybe, google earth tools will be useful
- it is no reported in this section how many specimens were collected from each area
- in which season samples were collected?
- it is no clear if authors measured principal water parameters at sampling moment
- line 81, which digital balance was used? (model, brand, City)
- line 81-82, which digital camera and which stereomicroscope were used (model, brand, City)
- information about image processing system/software are also absent
- Line 83-88, these statements are not clear, please rearrange, is there any reference supporting these statements?
- For shape analyses why authors did not used Shape R? "Libungan,L.A. & Pálsson,S. ShapeR: An R package to study otolith shape variation among fish populations. PLoS ONE 10, 1–12 (2015)
Results:
Results are generally well presented for the obtained data but are of difficult interpretation and application to the mean shape of asteriscii. I urge the authors to improve their results using shapeR to obtain the mean shapes of otolith contours for have a most comparable results between right and left asteriscii. Then, it would be helpful to perform a PCA on wavelet Fourier descriptors. Useful information and examples can be found in "Intra‐ and interspecific variability among congeneric Pagellus otoliths" (SciRep, 2021) and in "Otolith Analyses Highlight Morpho-Functional Differences of Three Species of Mullet (Mugilidae) from Transitional Water" (Sustainability, 2022). In "ShapeR: An R package to study otolith shape variation among fish populations" PLoS ONE 10, 1–12 (2015) it is possible to find the R procedures.
In presenting results in the table it is necessary to uniform the decimal separator, it should be the point (.).
Discussion:
It would be better to be more careful in assessing that pH and temperature influences the FA in the studied populations. To determine this statement it would be better to perform experimental procedures in which the only altered variables are pH and temperature. In fact, in natural condition FA can be ascribed to many factors, e.g. diet composition, currents velocity and others.
In conclusion section please avoid the use of statements as 241-244. In fact the authors cannot say that also other otoliths may show FA without present data about this statement.
All the best regards
The Reviewer
Author Response
Dear Reviewer 1 :
Please indicate species authorities each first time a species is mentioned, e.g. Cyprinus carpio (Linnaeus, 1758)
Done in the introduction and discussion sections and for other species (i.e. line 46…)
Otolith is a too general term, please use the specific term (asteriscus/ii) in indicating this structure
After the introduction where 3 different otoliths are presented, I added asteriscii before otolith as requested by reviewer
Introduction:
- lines 29-30 Otoliths are also use in taxonomy, and phylogeny, thus, are not primarily used for fisheries sciences
we modified the text by :
“Otoliths, calcified structures in the inner ears [1,2], are used to identify the species in taxonomic or phylogenic studies and to produce the necessary ageing data for management and assessment of stocks”
- please wide the introduction section regarding shape analyses ("Intra‐ and interspecific variability among congeneric Pagellus otoliths" (SciRep, 2021), "Otolith Analyses Highlight Morpho-Functional Differences of Three Species of Mullet (Mugilidae) from Transitional Water" (Sustainability, 2022), "ShapeR: An R package to study otolith shape variation among fish populations" PLoS ONE 10, 1–12 (2015)
We added the reference
D’Iglio, C.; Natale, S.; Albano, M.; Savoca, S.; Famulari, S.; Gervasi, C.; Lanteri, G.; Panarello, G.; Spanò, N.; Capillo, G. Otolith Analyses Highlight Morpho-Functional Differences of Three Species of Mullet (Mugilidae) from Transitional Water. Sustainability 2022, 14, 398. https://doi.org/10.3390/su14010398
we modified the text by :
The otoliths grow throughout the fish life and are metabolically inert [3], and their shape (i.e. their outline resulting from genetic, environmental and ontogenic effects) is used as an efficient tool to recognize the species at the interspecific level and to identify the fish stocks at the intraspecific level.
Material and methods:
- it is preferable to present "samples collection" before describing "study areas"
we modified the order and we added the morphometrical analysis
- in describing the sampling point please be sure to indicate geographical coordinates both in the text and in preparing the map reported in figure 1. Maybe, google earth tools will be useful
We used google earth and we added the geographical coordinates in the text :
“A total of 263 individual carp were collected from three different locations on the Euphrates (Nasiriya City; 31°2′28″ N, 46°14′45″ E), Tigris (Amarah City; 31°51′25″ N, 47°8′15″ E) and Shatt al-Arab Rivers (Basrah City; 30°33′41″ N, 47°47′41″ E) (Figure 1). “
- it is no reported in this section how many specimens were collected from each area
A total of 263 individual carp were collected from three different locations on the Euphrates (Nasiriya City; 31°2′28″ N, 46°14′45″ E; n=90), Tigris (Amarah City; 31°51′25″ N, 47°8′15″ E; n=89) and Shatt al-Arab Rivers (Basrah City; 30°33′41″ N, 47°47′41″ E; n=84) (Figure 1).
- in which season samples were collected?
- it is no clear if authors measured principal water parameters at sampling moment
- no, we used the available data from the same sampling area from the identified literature (15; 20; 26,27).
- line 81, which digital balance was used? (model, brand, City)
- line 81-82, which digital camera and which stereomicroscope were used (model, brand, City)
- information about image processing system/software are also absent
Images of the otolith were obtained directly on computer attached to the microscope and attached camera.
- Line 83-88, these statements are not clear, please rearrange, is there any reference supporting these statements?
We modified the text and we added the reference
Russ, J. Computer-Assisted Microscopy.-The Measurement and Analysis of Image, New York: Plenum Press Corp, 1990.
- For shape analyses why authors did not used Shape R? "Libungan,L.A. & Pálsson,S. ShapeR: An R package to study otolith shape variation among fish populations. PLoS ONE 10, 1–12 (2015)
We used only the univariate data as morphological and shape factors as many otolith shape papers. We don’t apply the Elliptic Fourier Descriptors (EFDs). If we applied the EFD analysis, we could use single R script (without ShapeR) to choice the Fourier Power because ShapeR is restricted the individual fourier power.
Results:
Results are generally well presented for the obtained data but are of difficult interpretation and application to the mean shape of asteriscii. I urge the authors to improve their results using shapeR to obtain the mean shapes of otolith contours for have a most comparable results between right and left asteriscii. Then, it would be helpful to perform a PCA on wavelet Fourier descriptors. Useful information and examples can be found in "Intra‐ and interspecific variability among congeneric Pagellus otoliths" (SciRep, 2021) and in "Otolith Analyses Highlight Morpho-Functional Differences of Three Species of Mullet (Mugilidae) from Transitional Water" (Sustainability, 2022). In "ShapeR: An R package to study otolith shape variation among fish populations" PLoS ONE 10, 1–12 (2015) it is possible to find the R procedures.
We used only the
In presenting results in the table it is necessary to uniform the decimal separator, it should be the point (.).
We modify all decimal separator by point.
Discussion:
It would be better to be more careful in assessing that pH and temperature influences the FA in the studied populations. To determine this statement it would be better to perform experimental procedures in which the only altered variables are pH and temperature. In fact, in natural condition FA can be ascribed to many factors, e.g. diet composition, currents velocity and others.
Yes, we are totally agree. This first in situ approach must be confirmed by experimental approach. There is a sentence in the conclusion section :
“the experimental approach should be used to better understand the factors controlling otolith shape integrating the ontogenic effect, as has been done for other species.”
In conclusion section please avoid the use of statements as 241-244. In fact the authors cannot say that also other otoliths may show FA without present data about this statement.
We deleted this sentence (l241-244).
All the best regards
Thank a lot for your comments
The Reviewer

Reviewer 2 Report
The current MS aims to study the potential asymmetry of otolith growth and shape variation in the common carp Cyprinus carpio by analysing right and left otoliths among 3 geographical distinct locations.
In general, the MS is relevant, considering a question that sometimes is not consider in otolith analysis. Furthermore, this work provides relevant information for future works. The manuscript is generally well written, with the objectives clearly described. It also uses adequate methods for the work proposed.
I would therefore recommend for publication in Fishes, provided the following issues be address / clearly explain in the MS.
Table 1: the minimum and maximum values for length, weight and width are usually equal for the 3 locations. Are these correct?
L81: The authors use the photographs to obtain length and width, so why not use the images to also calculate area and perimeter? Furthermore, author can also obtain EFDs and/or Wavelet descriptors. And why did the authors only calculate ellipticity and aspect-ratio?
L194: ontogeny can indeed interfere on otolith shape development, so why the author did not consider sex and sexual maturation in this work?
L201-205: Is there any kind of migration reported for the species? Migrations related to food, sexual development, or reproduction can influence the formation, and shape, of the otoliths.
L243: “… when otolith shape is used as tool for fish stock identification.” Not only for shape analysis, the asymmetry can also be a bias for microchemistry analysis, for instance.
Author Response
Dear Reviewer 2
The current MS aims to study the potential asymmetry of otolith growth and shape variation in the common carp Cyprinus carpio by analysing right and left otoliths among 3 geographical distinct locations.
In general, the MS is relevant, considering a question that sometimes is not consider in otolith analysis. Furthermore, this work provides relevant information for future works. The manuscript is generally well written, with the objectives clearly described. It also uses adequate methods for the work proposed.
I would therefore recommend for publication in Fishes, provided the following issues be address / clearly explain in the MS.
Table 1: the minimum and maximum values for length, weight and width are usually equal for the 3 locations. Are these correct?
Yes, we verified that in several cases, the value is the same from 3 areas but it is not the same individual. This table showed the value for all individuals for each area.
L81: The authors use the photographs to obtain length and width, so why not use the images to also calculate area and perimeter? Furthermore, author can also obtain EFDs and/or Wavelet descriptors. And why did the authors only calculate ellipticity and aspect-ratio?
We want to use only single measures as length, width and weight as several another studies and so with these three measures, only Ellipticity and Aspect ratio could be calculated. Of course, there are another approaches especially the Effiptic Fourier Descriptors (EFDs).
L194: ontogeny can indeed interfere on otolith shape development, so why the author did not consider sex and sexual maturation in this work?
It is not very easy to identify without bias the sex and maturity staging in these specimen from commercial catches.
We added “All individuals were from commercial catches and macroscopic observation of the gonads did not identify sex with a good accuracy.”
L201-205: Is there any kind of migration reported for the species? Migrations related to food, sexual development, or reproduction can influence the formation, and shape, of the otoliths.
This species in the Irak waters, does not migrate much.
l243: “… when otolith shape is used as tool for fish stock identification.” Not only for shape analysis, the asymmetry can also be a bias for microchemistry analysis, for instance.
Yes, we are totally agree but this sentence was deleted as requested by the reviewer 1.
Thank a lot for your comments

Reviewer 3 Report
Overall, I think the work could be useful. However, I find this manuscript needs extensive revision before it can be considered. The introduction does not set up the question, or especially the justification, very well. Why should we care about this? Readers need more detail than is given. What are the consequences of ignoring differences? Give concrete examples.
The results are poorly organized. The methods and results should be revised so they two parts reflect each other and are consistent. Consistent definitions of variables in the model, a full description of all models, and then the full model output, not just p-values, needs to be incorporated. It’s hardly surprising that you found statistical differences in otolith shapes based on size, etc. when you had a sample size that big. What matters is if the differences are big enough to change decisions regarding management of these fish stocks. Are they? Also, the intention was to show differences induced by environmental factors – why are those not visible in plots?
11: Fish age, not ageing
14-18: Please add how these measurements might affect the assessment of fish stocks. From what’s written, it’s unclear how changes in these shapes would affect further analyses.
20: “studied” not necessary in the sentence, add space between “area” and “and”
21: Again, I’m not seeing the relevance of otoliths becoming more elliptical through a fish’s life. How does this relate back to fish stock analyses? E.g. summarize the Mahé 2019 paper for those of use not familiar with these assessments.
30: delete “necessary” should just read “are used primarily to collect age data for management”
32-33: You must expand on this here, give specific examples of what happens from assuming.
39: I don’t think this is true, you’re referencing a study on a single species. All these use lapilli: https://doi.org/10.1111/eff.12598
https://doi.org/10.1002/tafs.10012
https://doi.org/10.2307/3536731
https://doi.org/10.1577/M07-115.1
https://doi.org/10.1111/jfb.12892
47: But there are more from around the world, right?
72: I can’t tell because of the page break, but this appears to be a single-sentence paragraph. Combine with the following (and indent). Consider putting numbers collected from each site on the map; also the map or placenames (or both) need to be larger to be legible
79: The first step was collecting and preserving the fish, you need to add this at least briefly, as some preservatives prevent the otoliths from being used (formalin, as is my understanding). Also, fish shrink after preservation, so adding how you dealt with changes in fish length would be useful. Perhaps the fish were frozen, or perhaps dissected immediately. Adding and entire paragraph on fish sampling, preservation, and otolith removal would not make the manuscript appreciably longer but would help with the replicability.
80: Just say “both asteriscii otoliths”
99: Why the log transform? Were the residuals normalized by the transform? Did the data span more than an order of magnitude? Also move this to later, we need to see the models first, not that you log-transformed the data first. The only variables I see preceding this are otolith length, weight, and width, which are used to calculate ellipticity and aspect ratio. What’s otolith feature? The variables defined earlier need to be exactly what’s in the model here, otherwise it’s hard to figure out what you’re modeling (or impossible). Things like “side effect” are obvious, but “geographical effect” is not, which appears in a Table, but in neither model presented here.
Eqn. (3) once you’ve clearly defined what these are, consistent with the preceding text, then I need to know how many of these were correlated – almost certainly all “features” are going to be highly correlated with otolith area, which means you’re dealing with multi-collinearity and potentially changing signs of parameter estimates depending on what variables are included in the model.
https://doi.org/10.2307/1937887
https://doi.org/10.1080/00031305.1982.10482818
https://doi.org/10.1002/wics.84
doi :10.1088/1742-6596/949/1/012009
107: “extracted from”
115-116: please provide the versions and citations for all packages used (you can use citation(“HH”) in R to get the citation, for example). The given citation is for a textbook, and better suited to justifying your choice of models, not the actual software. Give credit to the people that made the software and packages, not the people that wrote the book here.
119-120: This is a really uninteresting way to start a sentence. Give us results, not “look at the table,” especially as the opening paragraph. This sentence paves the way for the following sentences. The results overall are poorly written – what are we supposed to be looking at in Figure 3? Say what the differences are, and then use a reference to figure 3 to point it out. Otherwise you’re just saying look at figure 3.
123: dangling modifier here, who was using generalized linear models? Not the otolith descriptors. Also, what was the relationship? Spell these things out specifically, then reference the appropriate table/figure. Almost every sentence needs this correction, these are the results, tell us what they were, not just that “there were differences”
132-134: Difference in what?
142-typo in here, not sure what the sentence is supposed to say
Table 1: This might be more useful if it presented the difference between pairs, not the group means. As it is, there appears to be very little differences overall, but when paired there are differences?
Table 2 – seeing the actual estimates or effect sizes is far more useful that seeing p-values. Similarly, describing the effect sizes and if they are meaningful in the text is more useful that statistical significance.
Figure 4: “Value” on the y-axis is not appropriate. Value of what? Is it the title for each subpanel? If so, put that on the y-axis and get rid of the title. Whatever those regression lines are, they aren’t explained from the text as they are nonlinear.
236: Please make sure the major conclusions are readily visible in figures. There is no figure showing the linkage between asymmetry and geographical area, nor with shaped linked to environmental effects. There is a table of mean values showing differences between rivers but nothing showing the actual data for environmental covariates.
Author Response
Dear Reviewer 3
Overall, I think the work could be useful. However, I find this manuscript needs extensive revision before it can be considered. The introduction does not set up the question, or especially the justification, very well. Why should we care about this? Readers need more detail than is given. What are the consequences of ignoring differences? Give concrete examples.
We modified the end of the introduction section by :
“The present study focuses on potential asymmetry and the geographical effects on otoliths growth and shape in common carp Cyprinus carpio, a freshwater species of Iraq. A recent study using the otolith shape to identify the stock structure of the bogue (Boops boops; Linnaeus, 1758) in Mediterranean Sea showed that the significant asymmetry could modified the boundaries of stocks according to use the left or right otolith [15]. Moreover, this study showed that this significant asymmetry could be due to the environmental difference. Consequently, the aims of this study is to compare the fluctuations in asymmetry of asteriscus in the carp, Cyprinus carpio and to identify if this asymmetry could be a possible response to the environmental variables.in several rivers in Iraq.”
The results are poorly organized. The methods and results should be revised so they two parts reflect each other and are consistent. Consistent definitions of variables in the model, a full description of all models, and then the full model output, not just p-values, needs to be incorporated. It’s hardly surprising that you found statistical differences in otolith shapes based on size, etc. when you had a sample size that big. What matters is if the differences are big enough to change decisions regarding management of these fish stocks. Are they? Also, the intention was to show differences induced by environmental factors – why are those not visible in plots?
There are a mistake in the mean data of euphrate river and of course, we modified these information but another point is that the table 1 present the mean data by river, this is a good data but the efficient data are the individual data. Consequently, we sent the table 1 in the supplementary file. To show the differences among rivers/environment, we modified the Figure 3 in 3 parts/river.
11: Fish age, not ageing
Done
14-18: Please add how these measurements might affect the assessment of fish stocks. From what’s written, it’s unclear how changes in these shapes would affect further analyses.
We added :
“At one fish length, the size and/or the shape of otoliths could be different between two individuals and/or between left and right asteriscii otoliths for the same individual.”
20: “studied” not necessary in the sentence, add space between “area” and “and”
done
21: Again, I’m not seeing the relevance of otoliths becoming more elliptical through a fish’s life. How does this relate back to fish stock analyses? E.g. summarize the Mahé 2019 paper for those of use not familiar with these assessments.
We added :
“To use the otolith shape, it is essential to take into account the developmental stage of individuals to integrate the ontogenic effect”
30: delete “necessary” should just read “are used primarily to collect age data for management”
We modified :
“Otoliths, calcified structures in the inner ears [1,2], are used to identify the species in taxonomic or phylogenic studies and to collect age data for management and assessment of stocks.”
32-33: You must expand on this here, give specific examples of what happens from assuming.
We modified the text by ;
“The otoliths grow throughout the fish life and are metabolically inert [3], and their shape (i.e. their outline resulting from genetic, environmental and ontogenic effects) is used as an efficient tool to recognize the species at the interspecific level and to identify the fish stocks at the intraspecific level (to see the Stock Identification Methods Working Group of which identifies all studies each year; SIMWG)”
39: I don’t think this is true, you’re referencing a study on a single species. All these use lapilli: https://doi.org/10.1111/eff.12598
https://doi.org/10.1002/tafs.10012
https://doi.org/10.2307/3536731
https://doi.org/10.1577/M07-115.1
https://doi.org/10.1111/jfb.12892
Yes, it is not common for all freshwater species, We modified the text by :
“Conversely, asteriscus otoliths are the most frequently used otoliths in Cypriniforms species as common carp [6].”
47: But there are more from around the world, right?
Yes of course, we added “concerning the Iraq waters”
72: I can’t tell because of the page break, but this appears to be a single-sentence paragraph. Combine with the following (and indent). Consider putting numbers collected from each site on the map; also the map or placenames (or both) need to be larger to be legible
Done and we modified the Figure 1 with these good comments
79: The first step was collecting and preserving the fish, you need to add this at least briefly, as some preservatives prevent the otoliths from being used (formalin, as is my understanding). Also, fish shrink after preservation, so adding how you dealt with changes in fish length would be useful. Perhaps the fish were frozen, or perhaps dissected immediately. Adding and entire paragraph on fish sampling, preservation, and otolith removal would not make the manuscript appreciably longer but would help with the replicability.
We added some details ;
A total of 263 individual carp were collected in May-July 2021 from three different locations on the Euphrates (Nasiriya City; 31°2′28″ N, 46°14′45″ E; n=90), Tigris (Amarah City; 31°51′25″ N, 47°8′15″ E; n=89) and Shatt al-Arab Rivers (Basrah City; 30°33′41″ N, 47°47′41″ E; n=84) (Figure 1). All individuals were from commercial catches and macroscopic observation of the gonads did not identify sex with a good accuracy.
The commercial catches were sampled directly at the laboratory to limit the storage bias on the fish and otolith data.
80: Just say “both asteriscii otoliths”
Done
99: Why the log transform? Were the residuals normalized by the transform? Did the data span more than an order of magnitude? Also move this to later, we need to see the models first, not that you log-transformed the data first. The only variables I see preceding this are otolith length, weight, and width, which are used to calculate ellipticity and aspect ratio. What’s otolith feature? The variables defined earlier need to be exactly what’s in the model here, otherwise it’s hard to figure out what you’re modeling (or impossible). Things like “side effect” are obvious, but “geographical effect” is not, which appears in a Table, but in neither model presented here.
Yes, we don’t need to transform in the log because we used the residuals in the model. We deleted this sentence.
We clarify some points in the method. We modify area by river more clear to follow. We clarified the three Size parameters (otolith length, width and weight) and two shape indices (Ellipticity, Aspect-Ratio) and after the asteriscii otolith features (otolith length, width, weight, ellipticity and aspect-ratio).
Eqn. (3) once you’ve clearly defined what these are, consistent with the preceding text, then I need to know how many of these were correlated – almost certainly all “features” are going to be highly correlated with otolith area, which means you’re dealing with multi-collinearity and potentially changing signs of parameter estimates depending on what variables are included in the model.
https://doi.org/10.2307/1937887
https://doi.org/10.1080/00031305.1982.10482818
https://doi.org/10.1002/wics.84
doi :10.1088/1742-6596/949/1/012009
Yes, we verify for each model, the normality and the homoscedasticity of the residuals were assessed by visual inspection of diagnostic plots. Moreover, we added the sentence to be more clear : “One generalized linear model was performed by each otolith feature (otolith length, width, weight, ellipticity and aspect-ratio).”
Finally, we presented the data from each river in the figure 4 to see the orientation of asymmetry
107: “extracted from”
done
115-116: please provide the versions and citations for all packages used (you can use citation(“HH”) in R to get the citation, for example). The given citation is for a textbook, and better suited to justifying your choice of models, not the actual software. Give credit to the people that made the software and packages, not the people that wrote the book here.
Yes of course, we added all references :
“Statistical analyses were performed in the R statistical environment [30], using car [31] , sp [32], HH [33], vegan [34] and ggplot2 [35] packages.”
119-120: This is a really uninteresting way to start a sentence. Give us results, not “look at the table,” especially as the opening paragraph. This sentence paves the way for the following sentences. The results overall are poorly written – what are we supposed to be looking at in Figure 3? Say what the differences are, and then use a reference to figure 3 to point it out. Otherwise you’re just saying look at figure 3.
Yes we deleted the first sentences and we begin with the next sentence which explained that the right otolith was bigger than the left.
“There were some difference between left and right asteriscii otoliths from the same individuals with the right otolith was bigger (i.e. otolith length and width) than the left otolith (Figure 3).”
123: dangling modifier here, who was using generalized linear models? Not the otolith descriptors. Also, what was the relationship? Spell these things out specifically, then reference the appropriate table/figure. Almost every sentence needs this correction, these are the results, tell us what they were, not just that “there were differences”
We modified the first paragraph by “There were some difference between left and right asteriscii otoliths from the same individuals with the right otolith was bigger (i.e. otolith length and width) than the left otolith (Figure 3). Analysis of the relationships between fish length and each otolith feature (three size parameters and two shape indices) showed that there was a significant relationship between only two otolith size descriptors (length and width) with the total length of fish (column “TL”, Table 1). For three other otoliths features, the relationship with the total length was not significant (Figure 4). “
132-134: Difference in what?
This sentence is not clear and this table did not showed this point. We preferred to delete this sentence.
142-typo in here, not sure what the sentence is supposed to say
Yes it is not clear and we preferred to delete this sentence
Table 1: This might be more useful if it presented the difference between pairs, not the group means. As it is, there appears to be very little differences overall, but when paired there are differences?
Yes we are agreed and we sent this table in the supplementary table and we modified the data from the euphrate river because we presented the mistake .
Table 2 – seeing the actual estimates or effect sizes is far more useful that seeing p-values. Similarly, describing the effect sizes and if they are meaningful in the text is more useful that statistical significance.
We presented the p-value to show the significant effect and after with another figures, we characterized the effect.
Figure 4: “Value” on the y-axis is not appropriate. Value of what? Is it the title for each subpanel? If so, put that on the y-axis and get rid of the title. Whatever those regression lines are, they aren’t explained from the text as they are nonlinear.
We modified Values by Otolith measure and we presented the non linear lines to have a best view of the difference between left and right otoliths. We presented this figure by each river to see the environmental effect.
236: Please make sure the major conclusions are readily visible in figures. There is no figure showing the linkage between asymmetry and geographical area, nor with shaped linked to environmental effects. There is a table of mean values showing differences between rivers but nothing showing the actual data for environmental covariates.
We modified the figures with the comments of all reviewers to present a clear link between asymmetry and environment specially Figure 4
Thank a lot for your comments

Round 2
Reviewer 1 Report
The authors considered most of my comments improving the quality of the manuscript.
Despite this, in the response to my comments about results, authors leaved a truncated sentence: "We used only the". It would be better to present a detailed response to comments on results.
Please double check that latin terms are reported in italics.
Figure 4 is of low quality, please consider to present an high definition one.
Author Response
Dear reviewer 1
Reviewer 1
The authors considered most of my comments improving the quality of the manuscript.
Despite this, in the response to my comments about results, authors leaved a truncated sentence: "We used only the". It would be better to present a detailed response to comments on results.
We added a sentence in the results section :” To describe the otolith shape, only the univariate data as size parameters and shape factors were used.”
Please double check that latin terms are reported in italics.
Done, in results section, we modified the latin terms in italic
Figure 4 is of low quality, please consider to present an high definition one.
I changed the quality and I provided the image in tif format with high quality
thank a lot for your help

Reviewer 3 Report
No additional comments
Author Response
thank a lot for your help
